# Bidirectional Association between Periodontitis and Thyroid Disease: A Scoping Review

**DOI:** 10.3390/ijerph21070860

**Published:** 2024-06-30

**Authors:** Francesco Inchingolo, Angelo Michele Inchingolo, Alessio Danilo Inchingolo, Maria Celeste Fatone, Laura Ferrante, Pasquale Avantario, Arianna Fiore, Andrea Palermo, Tommaso Amenduni, Francesco Galante, Gianna Dipalma

**Affiliations:** 1Department of Interdisciplinary Medicine, University of Bari “Aldo Moro”, 70124 Bari, Italy or francesco.inchingolo@uniba.it (F.I.); or a.inchingolo3@studenti.uniba.it (A.M.I.); or laura.ferrante@uniba.it (L.F.); or pasquale.avantario@uniba.it (P.A.); or arianna.fiore@uniba.it (A.F.); or gianna.dipalma@uniba.it (G.D.); 2PTA Trani-ASL BT, Viale Padre Pio, 76125 Trani, Italy; tommaso.amenduni@aslbat.it (T.A.); francesco.galante@aslbat.it (F.G.); 3College of Medicine and Dentistry, Birmingham B4 6BN, UK; andrea.palermo2004@libero.it

**Keywords:** cytokines, hyperthyroidism, hypothyroidism, oral microbiome, periodontitis, scaling and root planning, thyroid cancer, thyroid disease

## Abstract

Periodontitis is a chronic inflammatory disease of the tissues surrounding and supporting the teeth. Due to the development of chronic inflammation, periodontitis can contribute to the development of several systemic diseases, including thyroid disease. Thyroid pathology includes benign, malignant, and autoimmune conditions leading to hypothyroidism, hyperthyroidism, or euthyroidism. Alterations in thyroid hormones, especially hypothyroidism, can reveal significant oral manifestations, including periodontitis. This scoping review aims to explore the probable causal relationship between periodontitis and thyroid disease, in terms of epidemiology, pathogenesis, and treatment. The search strategy follows the PRISMA-ScR guidelines. PubMed, Scopus, Web of Science, and Cochrane were searched from January 2014 to January 2024, entering the MESH terms “periodontitis” and “thyroid”. Of 153 initial records, 20 articles were selected and discussed. There is a high prevalence of periodontitis among patients with thyroid disease, including thyroid cancer. The causes at the basis of this association are genetic factors, the oral microbiome, and proinflammatory cytokines. Periodontal treatment, specifically scaling and root planning, can ameliorate thyroid parameters. Although there are a few randomized controlled studies in the literature, this review lays the foundation for a bidirectional relationship between periodontitis and thyroid disease, the link to which is, once again, systemic inflammation.

## 1. Introduction

Periodontitis is a chronic inflammatory disease of the tissues surrounding and supporting the teeth [1,2,3,4]. These tissues include the gums, the alveolar bone, and the periodontal ligament [5,6,7,8,9,10]. The disease originates from the accumulation of bacterial plaque and tartar on the teeth, which causes an inflammation of the gums known as gingivitis [11,12,13,14,15,16]. If untreated, gingivitis can involve deeper tissues, leading to periodontitis [17,18,19,20,21,22]. Far from being a pathology confined to the oral cavity, the implications of periodontitis can affect an individual’s systemic health [23,24,25,26,27]. In addition to local consequences such as tooth loss and damage to oral tissues, periodontitis can contribute to the development of several systemic diseases [28,29,30,31,32]. These associations derive from different pathogenetic mechanisms linking oral health to general health, primarily, and to systemic inflammation [33,34,35,36,37,38]. Systemic inflammation is associated with conditions such as cardiovascular diseases, type 2 diabetes mellitus, neurodegenerative disorders such as Alzheimer’s disease, and cancer [39,40,41,42]. Inflammation can damage the walls of blood vessels, increasing the risk of atherosclerosis and heart failure [43,44,45,46,47]. Bacteria associated with periodontitis, such as *Porphyromonas gingivalis*, can enter the bloodstream and spread to other sites in the body, contributing to neoplasms such as colorectal cancer [48,49,50,51,52,53,54]. Furthermore, periodontitis can alter the body’s immune response, influencing the production of specific antibodies that may be involved in several autoimmune diseases, such as rheumatoid arthritis [48,49,50,51,52,53]. Finally, chronic inflammation associated with periodontitis can affect glucose metabolism and insulin resistance, contributing to the etiopathogenesis of type 2 diabetes mellitus [28,42,54,55,56,57].

Thyroid pathology is varied and complex, including benign, malignant, and autoimmune conditions [58,59,60]. From a functional point of view, thyroid disease can lead to hypothyroidism, hyperthyroidism, or euthyroidism [61,62]. Hypothyroidism can derive from lymphocytic autoimmune thyroiditis (or Hashimoto’s thyroiditis), damage to thyroid tissue caused by surgery, radioactive iodine treatment, or thyroid ablation, drug-induced thyroiditis, and postpartum thyroiditis [63]. Hyperthyroidism can be caused by Graves’ disease, toxic nodules, or toxic multinodular goiter, drug-induced thyroiditis (such as amiodarone), subacute thyroiditis, postpartum thyroiditis, and factual hyperthyroidism [64,65,66]. Hashimoto’s thyroiditis is the most common autoimmune thyroid disease, characterized by chronic inflammatory infiltrates [66,67,68]. In addition, Graves’ disease represents one of the most frequent autoimmune diseases that affect the thyroid gland [61,62,69,70]. Thyroid carcinoma is the most common endocrine malignant neoplasm worldwide [71,72]. The most common subtype is papillary thyroid carcinoma, followed by follicular carcinoma, medullary carcinoma, and anaplastic carcinoma [73,74,75,76,77]. Both hypothyroidism and hyperthyroidism can reveal significant oral manifestations [78,79,80,81]. Hypothyroidism, characterized by reduced metabolic activity of the thyroid gland, can negatively influence periodontal health, causing delay in tooth eruption, alterations in taste, inflammation of the salivary glands, and compromising the health of periodontal tissues [78,79]. Conversely, hyperthyroidism, with its increased body metabolism, can increase susceptibility to tooth decay and periodontal disease and cause burning mouth syndrome [79,82,83].

As regards the specific relationship with periodontitis, hypothyroidism can contribute to systemic inflammation in the body, which in turn can affect the health of your gums [84,85,86]. Hypothyroid rats are characterized by an increased progression of periodontal disease [87,88,89]. A significant increase in the depth of the periodontal pockets and the loss of clinical attachment were found in patients with hypothyroidism compared to normal subjects [90,91]. Hypothyroidism can compromise the immune system, making the body more susceptible to infections, including gum infections associated with periodontitis [92,93]. In addition, hypothyroidism can affect bone metabolism, leading to a decrease in bone mineral density, an alteration of the bone remodeling process, and increasing the risk of bone loss associated with periodontitis [94,95,96,97]. Finally, hypothyroidism can affect blood flow, compromising the body’s ability to deliver nutrients and oxygen to the gum tissues [98,99]. Reduced blood flow can also impair the healing ability of the gums [90,100]. 

Nevertheless, regarding hyperthyroidism, no specific studies have been reported in the text that directly demonstrate its effect on the periodontal condition, except a few case reports and experimental models [91]. Generally, the elevated metabolism associated with hyperthyroidism may have possible implications for periodontal health [101,102,103] 

Some studies have shown that people with autoimmune thyroiditis, such as Hashimoto’s thyroiditis, may have an increased risk of developing periodontitis [104,105,106]. Once again, systemic inflammation associated with autoimmune conditions represents the “*trait union*” with periodontitis [104,107,108]. Likewise, the chronic inflammation associated with periodontitis could have systemic effects, potentially affecting the immune system and contributing to conditions such as thyroiditis. [109,110,111,112]. These mechanisms include the involvement of antinuclear antibodies (ANA), cellular apoptosis or programmed cell death, and the role of superantigens [113,114,115,116,117]. ANA may be involved in damaging thyroid cells, thus contributing to the development of the autoimmune disease [104,118,119]. In Hashimoto’s thyroiditis, deregulation of thyroid cell apoptosis may contribute to thyroid damage and disease progression [120,121,122]. Superantigens are bacterial toxins that can massively activate the immune system, leading to inflammation and tissue damage [59,123,124]. It has been suggested that exposure to superantigens derived from bacteria associated with periodontitis could also contribute to the development of Hashimoto’s thyroiditis, triggering an uncontrolled immune response [125,126,127]. 

However, it is important to note that while there are hypotheses that suggest a possible connection between thyroid disease and periodontitis, a confirmed direct correlation and the exact nature of this relationship remain unclear, and no randomized controlled trials (RCT) are reported in the literature. Further research is needed to fully understand the underlying mechanisms and possible bidirectional connection between periodontitis and thyroid disease, as well as to develop optimal management and treatment strategies for patients who may be affected by both conditions. 

This scoping review aims to explore the probable causal relationship between thyroid disease and periodontitis. Early recognition and timely intervention for both conditions could improve overall clinical outcomes. Patients with thyroid disorders may respond differently to treatments for periodontitis. Consequently, more aggressive or additional treatment protocols may be necessary to control inflammation and prevent the progression of periodontal disease in patients with thyroid disease. 

## 2. Materials and Methods

In consideration of the broad nature of the topic and the purpose of the research, finalized to summarize the evidence in the literature on the topic to inform future clinical practice, we have chosen to process the results and describe them qualitatively in a scoping review [128]. The search was performed following the method of Arksey and O’Malley (2005) and reported following the Preferred Reporting Items for Systematic Reviews and Meta-Analyses statement for reporting scoping reviews (PRISMA-ScR) [129,130,131].

### 2.1. PECO Question

The PECO framework is well suited to identifying the association between a specific condition and exposure to a risk factor of various types, in this case, the association between thyroid pathology and periodontitis [132,133].

P (population): patients with thyroid disease.

E (exposure): periodontitis or periodontal bacteria.

C (comparison): patients not exposed to periodontitis or periodontal bacteria.

O (outcome): association between thyroid disease and periodontitis.

### 2.2. Sources of Information and Search Strategy

The electronic databases PubMed, Scopus, Web of Science, and Cochrane were searched to find papers that matched our topic, dating from 1 January 2014, up to 17 January 2024. The Medical Subject Headings (MESH) terms entered in search engines were: “periodontitis” AND “thyroid” (Table 1).

### 2.3. Inclusion and Exclusion Criteria

The inclusion criteria were the following: (1) English language; (2) any type of observational study, that is, retrospective cohort, prospective cohort, case-control, cross-sectional, and randomized controlled trials (RCT); (3) open access; (4) articles concerning the association between periodontitis and thyroid diseases (hypothyroidism, hyperthyroidism, Hashimoto’s thyroiditis, Graves’ disease, thyroid nodules, and thyroid cancer); and (5) only human subjects of any age and gender. 

The exclusion criteria were the following: (1) other languages except English; (2) reviews, meta-analyses, case reports, and case series; (3) off-topic articles; (4) animal models; and (5) in vitro studies. By off-topic studies, we mean studies in which periodontitis and thyroid disease are mentioned but not related to the correlation between periodontitis and thyroid disease, for example, concerning other systemic diseases or other tumors, or studies that are limited to treating gingivitis. 

### 2.4. Screening and Data Charting

After eliminating duplicates manually, the title, abstract, and full text were screened against the inclusion and exclusion criteria. The screening of records occurs in two phases: the first consists of screening the title and abstract, and the second consists of reviewing the complete text. The primary reviewer (M.C.F.) performed 100% screening, and two additional reviewers (A.M.I. and A.D.I.) performed the screening, analyzing the full text. Doubts were resolved through discussion with a senior reviewer (F.I.). The electronic search results were exported to an Excel spreadsheet and downloaded into Zotero (version 6.0.15). Data graphing was performed by L.F., P.A., and A.F. 

## 3. Results

### 3.1. Selection of Sources of Evidence

A total of 153 records were identified using the keywords “periodontitis” AND “thyroid”. When applicable, the automatic filters entered were only in English, only clinical studies, only humans, no reviews, and free full text. The consulted databases were Pubmed (60), Scopus (57), Web of Science (29), and Cochrane (9). 

During the phase of screening, the inclusion and exclusion criteria were applied based on the analysis of the title and the abstract. Only studies that focused on the association between periodontitis and thyroid disease were included, as regards epidemiology, pathogenesis, and treatment. 

After screening, 105 articles were excluded by the analysis of title and abstract, leading to 48 records. Hence, duplicates (17) were manually removed, resulting in 31 reports assessed for eligibility. After eligibility, 20 studies were included in the final analysis. The process is summarized using a PRISMA-ScR flowchart (Figure 1) [129,134].

### 3.2. Characteristics of Sources of Evidence

The extracted data are presented in the form of tables to align with the objective and scope of this scoping review. The studies concerning the epidemiology of the association between periodontitis and thyroid disease are described in Table 2, while the studies concerning the common pathogenetic factors shared by periodontitis and thyroid disease are reported in Table 3, with a schematic description of the study design, the study sample, parameters evaluated, and results. Table 3 represents the studies concerning the effect of periodontal treatment on thyroid disease and the effect of thyroxine replacement on periodontitis, in terms of treatment, follow-up, parameter evaluations, and outcomes (Table 2, Table 3 and Table 4). 

### 3.3. Critical Appraisal within Sources of Evidence

A qualitative thematic analysis will be carried out to provide an overview of the literature on the topic. The results will be discussed based on practice and research, and a useful conclusion will be drawn for future studies in this field. 

### 3.4. Results of Individual Sources of Evidence

All relevant outcomes data for each source of evidence are reported in the column “results” in Table 2 and Table 3 and in the column “outcomes” in Table 4.

### 3.5. Synthesis of Results

According to four authors, patients with thyroid disease present worsening of periodontal indices [135,136,139,140]. Two authors exclude this correlation; even according to another author, patients with thyroid disease have less plaque accumulation [137,141,154]. Regarding the correlation between periodontitis and thyroid pathology in terms of risk, according to two authors, there is a greater incidence of thyroid cancer in patients with periodontitis, while according to two others, there is a greater incidence of benign pathology, specifically hypothyroidism and Hashimoto’s thyroiditis [138,142,143,144]. As regards the pathogenetic factors shared by periodontitis and thyroid disease, 4/5 records focus on hypothyroidism, which is associated with an increased frequency of *HLA-DRB1*03* and **04*, increased oral biodiversity, and overexpression of specific genes [145,147,148,149]. The treatment articles also deal mostly with hypothyroidism; two are about non-surgical periodontal treatment (NSPT), and one is about levothyroxine replacement therapy. One author addressed the effect of NSPT on the risk of thyroid cancer; another addressed the effect of atelocollagen injections on Hashimoto’s thyroiditis; and another, finally, addressed the effect of hyaluronic acid injections on dysthyroid patients in general [92,150,151,152,153,155].

## 4. Discussion

### 4.1. Summary of Evidence

#### 4.1.1. Epidemiology of the Association between Thyroid Disease and Periodontitis

The cross-sectional study by Kshirsagar M.M. et al. aimed to assess the oral health status and treatment requirements of individuals with thyroid dysfunction (hypothyroidism, hyperthyroidism, and goiter). The severity of dental caries status and periodontal destruction were greater in the 100 individuals with thyroid disease than in the control group, in terms of gingival bleeding, loss of attachment, and caries. The severity of periodontitis and dental caries was higher in women than in men, as regards missing teeth due to caries and crowns required. No statistical difference between dentition status, periodontal status, treatment need scores, and thyroid dysfunction subgroups was found with the ANOVA test. However, potential limitations of this study include the sample size, the selection bias, and the inability to establish causality in cross-sectional designs. The results of this study might have important practical implications for dental practice, for example, suspending all dental procedures in suspected thyroid cases, protecting the thyroid with a special collar when performing x-rays in dental practices, providing timely dental interventions in patients with thyroid disease, and, conversely, correcting impaired thyroid function as soon as possible in people with periodontitis [135]. 

Similarly, in a comparative cohort study, Song. E. et al. assessed the prevalence and severity of periodontitis in different thyroid function groups, including individuals with hypothyroidism, hyperthyroidism, and euthyroidism, to identify potential preventive and therapeutic strategies. In total, 5468 individuals were classified into three tertiles based on thyroid-stimulating hormone (TSH) levels, and periodontal status was measured using the community periodontal index. A significant difference was observed in the weighted prevalence of periodontitis according to TSH tertiles, with an inversely proportional relationship: in the first tertile, 26.5% of participants were affected by periodontitis, in the second tertile, 23.1%, and finally in the third tertile, 20.9%. These results were confirmed only in the subgroup of male participants, not among female participants. The impact of serum TSH levels on periodontitis was evaluated by multiple logistic regression analysis. In the model adjusted only for age and sex, the odds ratio (OR) for periodontitis was 1.34 for the first tertile of TSH vs. the third tertile. Despite adjustment for age, sex, BMI, smoking, alcohol, exercise, fasting blood glucose, systolic blood pressure, total cholesterol, eGFR, AST, and ALT, the statistically significant association persists, with an OR of 1.39. Further adjustment for log-transformed urine iodine levels and thyroid peroxidase antibodies revealed significantly higher odds of periodontitis among subjects in the tertile group with lower TSH than those in the third tertile (OR 1.36). Nevertheless, the study results should be interpreted with caution due to potential limitations, such as data accuracy, selection bias, and the inability to establish causality in observational studies [144]. 

In addition, in a consistent retrospective study, Chrysanthakopoulos N.A. et al. found that the presence of deep periodontal pockets (>5 mm) was significantly associated with thyroid disease, as well as with male sex, smoking habit, congenital heart disease, vascular disease, mitral valve prolapse, hypertension, stroke, heart attack, other endocrine diseases, liver disease, diabetes mellitus, kidney disease, respiratory allergies, and anemia. Conversely, there was no association between thyroid disease and clinical attachment loss (CAL). According to the authors, periodontitis can be considered a risk factor for a systemic disease, including thyroid disease. However, the study has limitations, including self-reported medical histories and a cross-sectional design [136]. 

Specifically, Gao Y. et al. investigated the potential bidirectional relationship between periodontitis and thyroid function using Mendelian randomization, a methodological approach that uses genetic variants as instrumental variables to assess causal relationships between exposures and outcomes. The authors revealed a positive causal effect of genetically predicted periodontitis on the risk of hypothyroidism but did not demonstrate the inverse relationship (OR = 1.24). In contrast, no statistically significant association was found between periodontitis, hyperthyroidism, and autoimmune thyroiditis. This study helps to better clarify the nature of the association between periodontitis and thyroid disease, revealing that it is not alterations in thyroid function that cause periodontitis but rather periodontitis that causes hypothyroidism. These findings should be interpreted cautiously, considering potential limitations such as the assumptions of Mendelian randomization and the need for replication in independent cohorts. Factors such as lifestyle habits, socioeconomic status, and environmental factors may also influence periodontal health and thyroid function [142]. 

In contrast to the previous findings, several authors excluded a significant association between periodontal disease and thyroid diseases. Kwon M. et al. conducted a consistent population-based cross-sectional study, assessing oral health parameters such as periodontal health, dental caries, and oral hygiene habits in 18.034 adult patients. Paradoxically, histories of thyroid diseases were found to be more common in people who brush their teeth frequently or use oral hygiene products. This result could be explained because the female sex, to which better oral hygiene and a higher frequency of thyroid disease are related, acts as a confounding factor. However, the authors found significant associations between community periodontal indexes (CPI) ≥ 3 and abnormal thyroid function tests, even if there was no association between CPI and a history of thyroid disease. Higher CPI values revealed higher probabilities of abnormal thyroid tests [137]. 

Similar results are derived from Chatzopoulos, G.S. et al., who explored the complex relationship between periodontitis, systemic diseases, and smoking habits in a consistent retrospective study. The authors aim to understand the impact of periodontitis on systemic health outcomes and the influence of smoking on periodontal disease progression. Analyzing the final 2069 records, no statistically significant association was found between periodontal disease and thyroid disease. In contrast, multiple sclerosis and smoking were significantly associated with grade C periodontitis [141]. 

The comparative cohort study by Venkatesh Babu N. et al. examined the oral health status of children diagnosed with thyroid disorders. The study uses a cross-sectional design to assess parameters such as dental caries, periodontal health, oral hygiene practices, and treatment needs. Plaque and gingival scores were significantly higher in the thyroid group compared to the control group (healthy children), were the modified developmental defects of enamel score (DDE). The decayed missing filled teeth index for permanent teeth (DMFT) and decayed missing filled teeth index for primary teeth (dmft) scores were higher in the thyroid group than the control group, but without reaching the level of statistical significance. Moreover, children with thyroid disorders presented other oral manifestations such as macroglossia, an open bite, and a change in eruption pattern. The findings could have significant implications for pediatric dentistry and healthcare, highlighting the need for tailored oral health interventions and raising awareness among healthcare providers and parents about monitoring and addressing oral health issues in children with thyroid disorders. However, potential limitations of the study include the small sample size and the cross-sectional design [139]. 

In another cross-sectional study, Zeigler C.C. et al. focused on the link between periodontal health and blood pressure levels in obese adolescents, detecting an increase in TSH in patients with periodontal pockets (PD) ≥ 4 mm. These findings suggest a significant association between pathological PD and raised diastolic blood pressure in obese adolescents, highlighting the potential role of periodontal health in cardiovascular health outcomes. Although there are limitations related to its cross-sectional design, these data could have implications for preventive healthcare strategies and interventions aimed at reducing cardiovascular risk in obese adolescents. Future research should also consider factors such as diet, physical activity levels, and genetic predispositions [140]. 

Some authors focused on the relationship between periodontitis and thyroid cancer [156,157,158,159]. In general, periodontal disease is associated with the risk of developing a malignancy, particularly breast cancer in women, prostate cancer in men, hematologic malignancies, and gastrointestinal cancer [160,161,162,163,164,165]. 

The retrospective nationwide population-based cohort study by Kim, E.H. et al. explored the potential link between periodontal disease and cancer risk. The cumulative incidence of cancer in the group with periodontitis was 2.2 times higher than in the control group (patients without periodontitis), after adjustment for age, sex, comorbidities, BMI, and smoking history. In particular, the authors confirmed that periodontitis was statistically significantly associated with thyroid cancer with *adjusted hazard ratios* (aHR) of 1.307, as well as stomach cancer, colon cancer, lung cancer, bladder cancer, and leukemia [143]. 

In addition, the large comparative cohort study by Chen S.H. described the relationship between periodontitis, anti-periodontitis therapy, and extra-oral cancer risk. The study uses a nationwide population-based approach, leveraging a large dataset of 683.854 participants to understand the potential impact of periodontitis and its treatment on the risk of developing extra-oral cancers. In general, patients with periodontitis had a lower risk of cancer than healthy controls, regardless of gender, age, comorbidity, or use of nonsteroidal anti-inflammatory drugs. Probable treatments for periodontitis, including scaling and root planning (SRP) and periodontal flap surgery, might contribute to the reduction of cancer risk. Intriguingly, the exceptions to this finding are thyroid cancer, with an aHR = 1.20 for women and aHR = 1.51 for men, as well as prostate cancer. These findings should be approached with caution due to the potential limitations of this study, such as the data accuracy, the selection bias, and the inability to establish causality in observational studies [138]. 

For some types of neoplasms, including thyroid cancer, periodontitis could represent a modifiable risk factor, and SRP should be included in the management protocols of cancer patients. 

#### 4.1.2. Pathogenesis of the Association between Thyroid Disease and Periodontitis

The understanding of the pathogenetic mechanisms underlying the relationship between periodontitis and thyroid disease is crucial to discovering new therapeutic targets for both diseases. The pathogenetic mechanisms resulting from this scoping review are as follows: genetic factors, oral microbiome, and inflammatory cytokines (Figure 2).

#### 4.1.3. Genetic Factors

Human leukocyte antigen (HLA) genes encode proteins essential to the immune system’s self-self and non-self-discrimination. T cells are exposed to peptide antigens by the HLA-DR molecules, which are a subset of major histocompatibility complex (MHC) class II molecules. The different frequency of HDL HLA alleles predicts the immune response to a variety of systemic diseases of infectious, autoimmune, neoplastic, and genetic etiology.

Al-Hindawi S. et al. investigated a possible genetic link between periodontitis and hypothyroidism. A DNA polymerase chain reaction was performed on the genomes of patients with hypothyroidism, patients with hypothyroidism and periodontitis, and healthy controls to perform HLA genomic typing. The statistical analysis revealed that the frequencies of the *HLA-DRB1*03* and **04* alleles were significantly increased in hypothyroid patients with or without periodontitis compared with healthy controls. Conversely, the frequencies of the *HLA-DRB1*08* allele were significantly higher in the healthy group. The *HLA DRB1*03* allele was overexpressed in individuals with both hypothyroidism and periodontitis, suggesting a common genetic basis conferring susceptibility to both diseases. These alleles might make it possible for autoantigenic peptides to attach to T cells and be presented to them, which could lead to an autoimmune reaction. Furthermore, thyroid autoantigen presentation to T cells, which can result in autoimmune thyroid disorders like Hashimoto’s thyroiditis, is significantly influenced by *HLA-DRB1* alleles [145].

Zeng Y. demonstrated the effect of TSH on the osteogenic differentiation of the periodontal ligament. The periodontal ligament, which is part of the periodontium, is the connective tissue between the tooth root and alveolar bone, allowing elasticity to distribute chewing forces over a large surface area of the alveolar process. Periodontal tissue regeneration underlies many oral diseases, including periodontitis, malocclusion, and dental defects. Osteoblast formation from mesenchymal stem cells (MSCs) is dependent on extracellular matrix breakdown-related components. PLCG2 starts store-controlled calcium channel function. It helps to promote mesenchymal osteogenic differentiation by triggering store-operated calcium channel activation. The addition of TSH to a culture medium of human periodontal ligament stem cells (PDLSCs) resulted in a marked reduction in their osteogenic differentiation, decreasing calcium nodules, alkaline phosphatase levels, and collagen synthesis. TSH could alter the gene expression of some osteogenesis-related genes in the periodontal ligament. After adding TSH at various levels, the expression of *osteogenesis-related-PDLSC genes decreased*, namely *osteopontin (OPN)*, *RUNX family transcription factor 2 (RUNX2)*, *collagen type I alpha 1 chain (COL1A1)*, and *osteocalcin (OCN)*, in a directly proportional manner to TSH concentration. After under-recruited high-throughput sequencing analysis, some genes were found to be downregulated, including the *matrix metallopeptidase 3 (MMP3)*, *lymphocyte cytosolic protein 1 (LCP1)*, *and 1-phosphatidylinositol-4,5-bisphosphate phosphodiesterase gamma-2 (PLCG2)* genes. They are responsible for extracellular matrix degradation, activation of calcium channel activity, and interleukin (IL)-17 and tumor necrosis factor (TNF) signal transduction pathways, which regulate bone and periodontal remodeling. This finding has important therapeutic implications, so it is important to screen for and correct any hypothyroidism in patients with periodontitis and, conversely, to perform oral examinations in patients with hypothyroidism to recognize periodontitis [147].

Moreover, Liu, W. et al. revealed that thyroid cancer suppressors might play a role in the pathogenesis of periodontitis, acting on *PDLSCs* [146]. *Candidate papillary thyroid carcinoma susceptibility gene 3 (PTCSC3)* is a suppressor of thyroid cancer and glioma [166]. Toll-like receptor 4 (TLR4) is a transmembrane protein capable of increasing the expression of inflammatory cytokines that in turn activate the innate immune system [10,167,168]. It is overexpressed in human PDLSCs, partly because of hyperglycemia and insulin resistance [169]. PTCSC3 is a regulatory molecule that potentially modifies inflammatory responses in periodontitis by downregulating TLR4. PTCSC3 has been proposed to downregulate inflammatory pathways and prevent the growth of periodontal ligament stem cells (PDLSCs). Periodontitis-affected PDLSC proliferation was suppressed by overexpression of PTCSC3, which also resulted in the downregulation of TLR4. TLR4 plays a role in the immunological response associated with periodontitis by stimulating the release of inflammatory cytokines in response to bacterial infections. In the setting of periodontitis, overexpression of TLR4 has been connected to diseases, including insulin resistance and glucose intolerance, which are both linked to heightened inflammatory responses. On the other hand, PDLSC proliferation was unaffected by TLR4 overexpression. PTCSC3 and TLR4 were dysregulated in individuals affected by periodontitis. PTCSC3 was significantly downregulated, and TLR4 was significantly upregulated in PDLSCs from the teeth of individuals with periodontitis, respectively, but not from the teeth of healthy subjects. Increased expression levels of PTCSC3 can improve periodontitis by inhibiting PDLSC levels and reducing TLR4 levels [146]. 

#### 4.1.4. Oral Microbiome

Five articles deal with the oral microbiome, highlighting its significance in both oral and systemic health [43,148,170,171,172].

The relationship between oral microbiome and thyroid pathology represents a developing area of research because of the few studies. The oral microbiome consists of various microorganisms living in the mouth, playing a crucial role in both oral and systemic health. Imbalances in this microbiome can lead to oral diseases, such as periodontitis and caries, and are increasingly being linked to systemic conditions, including thyroid diseases [148]. Thyroid hormone alterations, such as hypothyroidism and hyperthyroidism, have been shown to have potential connections with the oral microbiome. Periodontal pathogens, for instance, may have systemic effects that could disrupt endocrine function, including the thyroid. The exact mechanisms are still being studied, but the presence of a dysbiotic oral microbiome (an imbalance of harmful and beneficial microorganisms) is believed to contribute to the inflammatory burden on the body, which can affect thyroid health [172]. Obviously, maintaining good oral hygiene and addressing oral infections promptly may help mitigate these risks. As research progresses, a clearer understanding of the oral microbiome’s role in thyroid pathology could lead to better preventive and therapeutic strategies for managing thyroid diseases. While the connection between the oral microbiome and thyroid pathology is still being explored, existing evidence suggests that oral health significantly impacts thyroid function, emphasizing the importance of maintaining a healthy oral microbiome for overall systemic health [171]. 

Alterations of the oral and intestinal microbiome are at the center of attention in the latest studies on the etiopathogenesis of numerous systemic diseases [172,173,174,175]. Zheng L. et al. demonstrated the reciprocal influences between thyroid hormones and the oral microbiome, to which periodontitis is related. Both clinical and subclinical hyperthyroidism are associated with reduced diversity of the oral microbiome on nonmetric multidimensional scaling graphs. Conversely, the presence of anti-thyroid oxidase antibodies and hypothyroidism are associated with increased oral biodiversity. No differences in the qualitative composition of the oral microbiome between the triiodothyronine, thyroxine, and tireoglobuline groups were detected. This study presents numerous limitations, linked to its transversal nature and the fact that the oral microbiome can be altered by numerous exogenous and endogenous factors, such as food and drink, systemic diseases, immune status, and drugs. Further studies are needed to understand the pathogenetic mechanisms by which thyroid hormones affect the oral microbiome and the specific types of bacteria involved [148].

#### 4.1.5. Pro-Inflammatory Cytokines

Inflammatory cytokine levels play a crucial role in both periodontitis and thyroid disease [176,177]. In an analytical cross-sectional study, Sahar H. Al-Hindawi et al. examined the cytokine levels in 60 hypothyroid patients, half of whom had periodontitis and the other half did not. Hypothyroid patients, particularly those with periodontitis, had considerably greater serum levels of IL-1β than the control group (healthy individuals). Patients with hypothyroidism, especially those who had periodontitis, also showed a significant increase in salivary IL-1β levels, which may indicate a biomarker for the disease. Serum IL-6 levels across patients’ groups and controls did not significantly differ, in contrast to IL-1β. However, compared to controls and non-periodontitis patients, salivary IL-6 levels were considerably higher in individuals with periodontitis, suggesting local inflammation. There were no discernible variations in TNF-α levels in serum or saliva between the study groups and the controls.

These results contribute to improving our knowledge about the function of cytokines in periodontitis and hypothyroidism. While elevated salivary IL-6 levels may indicate local inflammation in periodontitis, elevated IL-1β levels suggest its participation in thyroid pathophysiology and as a potential marker for the disease. Nevertheless, even though TNF-α may be a sign of periodontitis, its function in thyroid disorders is still unknown [149]. 

#### 4.1.6. The Effect of Periodontal Treatment on Thyroid Disease and the Effect of Thyroxine Replacement on Periodontitis

The most widespread non-surgical periodontal treatment consists of SRP associated with good oral hygiene [178,179,180,181,182]. Decreasing inflammatory cytokine levels and NSPT can reduce the degree of flogosis and oxidative stress. In a comparative study, Vallabhan C.G.et al. revealed that scaling and root planning (SRP) affects serum levels of TNF and IL-6 in patients with periodontitis and hypothyroidism and patients with periodontitis without hypothyroidism. At the periodontal level, TNF and IL-6 activate metalloproteinase, enzymes that destroy connective tissue, and activate osteoclasts, the cells responsible for bone destruction [183,184]. In hypothyroid patients, bone resorption is slower due to reduced metabolism [185,186]. Baseline measures of periodontal parameters, namely plaque index (PI), gingival index (GI), probing pocket depth (PPD), and clinical attachment level (CAL), were noted for both groups at baseline and 4 weeks after SRP. All clinical indicators were significantly reduced in both groups after SRP. Furthermore, following treatment, serum levels of TNF and IL-6 dramatically dropped. This research explains how the regulation of inflammatory processes and microbial load may be responsible for the reduction in serum cytokine levels that occur after NSPT [150]. 

In another small prospective interventional study, Bhankhar R. et al. evaluated serum TSH levels at baseline at 3 months after NSPT in hypothyroid patients and a control group. The mean TSH values in the hypothyroid patients were significantly reduced at 3 months. Clinical parameters improved in both groups. In addition, greater bone loss was found in hypothyroid patients [151]. 

Surprisingly mixed results derived from the retrospective cohort conducted by Hwang I.M. et al. In general, the risk of developing a malignancy was significantly lower in the group of patients undergoing NSPT (aHR = 0.72), specifically as regards malignancies of the gastrointestinal tract, lung, gynecological, and brain. As for thyroid cancer, along with prostate cancer, the risk was paradoxically higher in the group undergoing SRP (aHR = 1.54). Higher cancer screenings linked to more regular dental care may be the cause of the higher risk of thyroid and prostate malignancies after PD treatment. It can be postulated that casual discoveries made during normal dental exams contributed to the higher diagnosis of thyroid cancer in the PD therapy group [152]. 

In conclusion, from the analysis of the abovementioned studies, we infer that NSPT, by reducing the levels of inflammatory cytokines, could reduce the risk of neoplasia and restore TSH levels. The limitations of these studies are the small sample number and comparative nature.

According to Yusubova Sh.R. et al., patients with hypothyroidism and periodontitis may also benefit from using hyaluronic acid to treat periodontitis [153,187]. Hyaluronic acid is a glycosaminoglycan that is part of human and mammalian connective tissues [187,188,189,190]. Because of its immune-stimulating and correction properties, hyaluronic acid is being investigated as a potential therapeutic agent to enhance periodontal health [187,191,192]. In the first study group, patients underwent injection of hyaluronic acid into the periodontal tissues; in the second comparison group, patients used a common chlorhexidine antiseptic; and in the third control group, dental plaque removal, closed curettage, and application of a chlorhexidine solution were performed. Levels of pro-inflammatory and antioxidant enzymes, namely, malondialdehyde (MDA), superoxide dismutase (SOD), and catalase, were assessed by biochemical analysis of oral fluid. Levels of immunoglobulin IgG, IgM, and IgA in saliva. Periodontal and gingival indexes were worse in hypothyroid patients than in healthy controls. Significant improvements in biochemical markers and periodontal indexes, such as decreased levels of malondialdehyde and elevated antioxidant enzyme activity, were observed following hyaluronic acid treatment. Moreover, immunoglobulin levels rose after hyaluronic acid therapy [153]. 

Only the transversal comparative study by Rahangdale, S. et al. evaluated the effect of hormone replacement treatment on periodontal status. Periodontal parameters (PI, BOP, PPD, and CAL) and radiographic parameters (mandibular cortical width and panoramic mandibular index) were evaluated in a group of hypothyroid patients treated with thyroxine and compared with a control group of healthy patients. The hypothyroid patients had significantly higher PPD and CAL than the control group. No significant differences were observed with varying doses and durations of thyroxine treatment, as a link between dose and duration of therapy and periodontal status was ruled out. The limitations of this study are related to its cross-sectional nature and the small sample size. Thyroxine causes an increase in bone turnover with a decrease in bone density at the level of the periodontium; therefore, it would be appropriate to look for periodontitis in hypothyroid patients undergoing replacement therapy [92]. 

There is currently a lack of studies in the literature evaluating the effect of tapazole, a thyroperoxidase inhibitor, on periodontium in hyperthyroid patients.

### 4.2. Limitations of the Studies

The limitations of the studies examined are the following: the cross-sectional nature, the lack of RCTs, the small sample sizes, the selection bias, the parameter heterogeneity, and the selection of open-access articles. Despite conflicting data, this review provides insights for clinical practice, suggesting that endocrinologists should refer thyroid patients to dentists for periodontitis screening and, conversely, that dentists should pay special attention to thyroid patients, particularly during radiographic exams. Consistent RCTs are needed to clarify the relationship between thyroid disease and periodontitis.

## 5. Conclusions

This is the first comprehensive review of the relationship between thyroid disease and periodontitis. Some studies suggest patients with thyroid dysfunction, especially hypothyroidism, have a higher prevalence and severity of periodontal disease, possibly because of the effect of TSH on bone turnover. Other authors excluded a link between periodontitis and thyroid dysfunction, with some paradoxical data, indicating a higher prevalence of thyroid disease in those with better oral hygiene. Genetic factors, alterations in the oral microbiome, and inflammatory cytokines may contribute to the relationship between periodontitis and thyroid disease. Specifically, the *HLA-DRB1*08 allele*, dysregulation of genes related to periodontal ligament remodeling, and elevated levels of IL-1β and IL-6 have been noted in patients with altered TSH. Hyperthyroidism is associated with reduced oral microbiome diversity, while hypothyroidism is linked to increased diversity. More research is needed to understand the bidirectional influence between thyroid hormones and the oral microbiome. NSPT, like SRP, has shown promise in reducing inflammatory cytokines (IL-6 and TNF) and normalizing TSH levels in hypothyroid patients. Hyaluronic acid and atelocollagen injections also demonstrate benefits for hypothyroid and autoimmune thyroiditis patients, respectively. Only one study has examined the impact of thyroid therapy (thyroxine) on periodontium, finding it worsened periodontal parameters in hypothyroid patients. Further research should evaluate the effects of other thyroid medications on periodontal health. 

## Figures and Tables

**Figure 1 ijerph-21-00860-f001:**
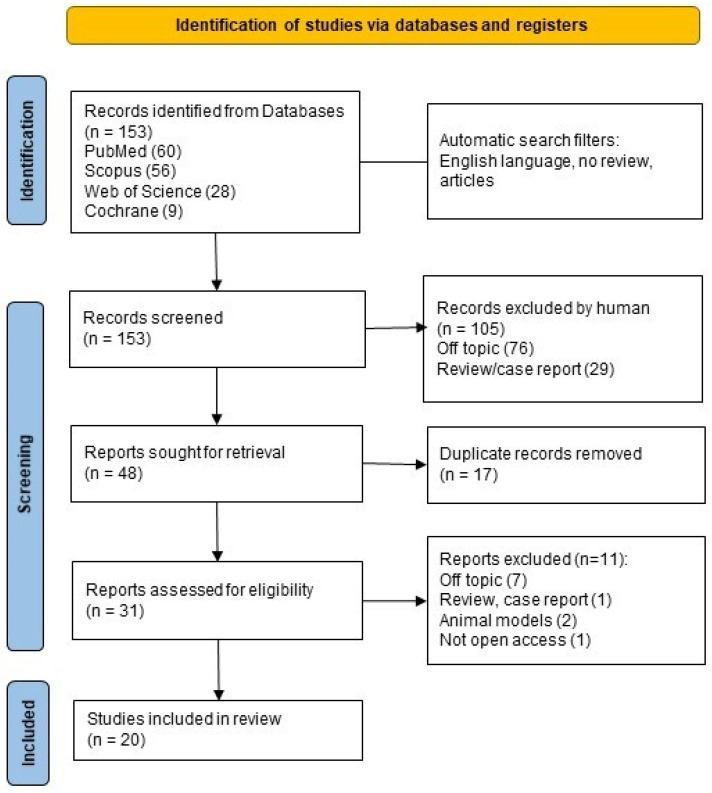
Preferred Reporting Items for Systematic Reviews and Meta-Analyses statement for reporting scoping reviews (PRISMA-ScR) flow-chart [129].

**Figure 2 ijerph-21-00860-f002:**
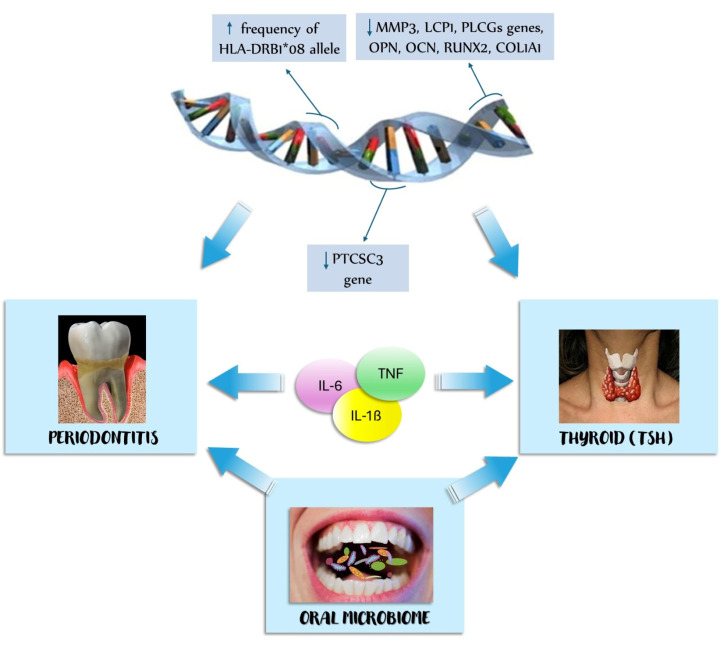
Pathogenetic mechanisms of the relationship between periodontitis and thyroid disease. COL1A1: collagen type I alpha 1 chain, HLA: human leukocyte antigen, IL-1: interleukine 1, IL-6: interleukine 6, LCP1: lymphocyte cytosolic protein 1, MMP3: matrix metallopeptidase 3, OCN: osteocalcin, OPN: osteopontin, PLCG2: 1-phosphatidylinositol-4,5-bisphosphate phosphodiesterase gamma-2, PTCSC3: candidate papillary thyroid carcinoma susceptibility gene 3, RUNX2: family transcription factor 2, TNF: tumor necrosis factor, TSH: thyroid-stimulating hormone.

**Table 1 ijerph-21-00860-t001:** Article screening strategy.

Articles Screening strategy	KEYWORDS: A: “periodontitis”; B: “thyroid”
	Boolean Indicators: “A” AND “B”
	Timespan: from 1 January 2014, to January 2024
	Electronic Databases: Pubmed, Scopus, Web of Science, and Cochrane

**Table 2 ijerph-21-00860-t002:** Studies concerning the epidemiology of the bidirectional association between periodontitis and thyroid disease.

Authors	Study Design	Study Sample (n. Patients)	Parameters Evaluated	Results
Kshirsagar et al., 2018 [135]	Cross-sectional	200 participants Study group (thyroid disease and periodontitis): 100 Control group (healthy): 100	Gingival bleeding, loss of attachment, and caries	Greater severity of dental caries status and periodontal destruction in patients with thyroid disease and periodontitis, especially in women
Chrysanthakopoulos et al., 2016 [136]	Retrospective	3.360 outpatients	Systemic diseases as independent variables and the relative frequency of periodontal pockets measuring ≥5 mm and CAL measuring ≥6 mm as dependent variables	Male gender, vascular illness, hypertension, stroke, heart attack, diabetes mellitus, endocrine disorders, thyroid disease, respiratory allergies, and rheumatoid arthritis are associated with periodontal pocket depth. CAL was linked to the illnesses, as well as infective endocarditis and chronic obstructive pulmonary disease, but not to other endocrine or thyroid disorders
Kwon et al., 2022 [137]	Cross-sectional	18.034 adults	Oral health-related parameters and the prevalence of thyroid diseases; oral health status, and TFTs	Thyroid disease histories are more prevalent in those who regularly clean their teeth or use mouthwash. CPI and a history of periodontitis do not significantly correlate with thyroid illness. In individuals without a prior history of thyroid diseases, the association of higher CPI with abnormal TFTs
Chen et al., 2023 [138]	Comparative cohort	683.854 participants	aHR for cancer aHR for specific types of cancer regardless of gender, age, Charlson comorbidity index, and the use of nonsteroidal anti-inflammatory drugs	Lower aHR for cancer in periodontitis patients Increased aHR for prostate cancer in men and thyroid cancer (aHR = 1.20 for women and aHR = 1.51 for men) in periodontitis patients Periodontal treatment reduced the risk of cancer
Venkatesh Babu et al., 2016 [139]	Comparative cohort	200 children Study group: 100 with thyroid dysfunction Control group: 100 healthy	GI, PI, DMFT, Dmft, and modified DDE index	Higher GI, PI, and DDE indexes in the thyroid group Higher DMFT and dmft index in the thyroid group, but this difference was not statistically significant Children with thyroid disease have additional oral symptoms (macroglossia, open bites, and altered eruption patterns)
Zeigler et al., 2015 [140]	Cross-sectional pilot	75 obese adolescents with PD < 4 mm: 61 PD ≥ 4 mm: 14	VPI, BOP, and PD ≥ 4 mm Systolic and diastolic blood pressures	Adolescents with PD ≥ 4 mm had significantly higher BOP, higher diastolic blood pressure, and higher serum levels of IL-6, Leptin, MCP-1, and TSH
Chatzopoulos et al., 2023 [141]	Retrospective	2.069 records	Grade of periodontitis	Men have widespread periodontitis, and older adults have grade B and IV periodontitis. Those with generalized disease and stage IV had greater tooth loss rates. Multiple sclerosis and smoking have been associated with grade C PD. No association between stage and extension of periodontitis and thyroid disorders (*p* value > 0.99).
Gao et al., 2024 [142]	Bidirectional, univariable MR framework	17.353 cases 28.210 controls	OR for periodontitis, FT4, TSH levels, hypothyroidism, hyperthyroidism, and AT	Increased OR (1.24) for hypothyroidism in individuals with a genetic predisposition to periodontitis
Kim et al., 2022 [143]	Retrospective cohort	713.201 individuals Periodontitis group: 53.075 Control group: 660.126	Cumulative incidence of cancer, aHR for specific types of cancer	Increased cumulative incidence of cancer (2.2) for cancer in the periodontitis group; higher aHR (1.307) for cancer in the periodontitis group adjusted for age, sex, comorbidities, BMI, and smoking history Increased aHR for leukemia, stomach cancer, colon cancer, lung cancer, bladder cancer, and thyroid cancer in the periodontitis group No significant association between the development of secondary malignancy and periodontitis in cancer survivors at 5 years
Song et al., 2021 [144]	Comparative cohort	5468 participants, 1423 with periodontitis First terzile: TSH < 1.76 mIU/L Second terzile: TSH 1.76–2.83 mIU/L Third terzile: TSH > 2.83 mIU/L)	CPI	OR for periodontitis is 1.34 in the first tertile vs. the third tertile, adjusted OR for risk factors is 1.39, and adjusted OR for log-transformed urine iodine levels and thyroid peroxidase antibodies is 1.36

API: approximal plaque index, AT: autoimmune thyroiditis, BOP: bleeding of probing, BMI: body max index, CPI: community periodontal index, DMFT: decayed missing filled teeth index for permanent teeth, Dmft index: decayed missing filled teeth index for primary teeth, DDE: developmental defects of enamel, GI: gingival index, IL-6: interleukin 6, MCP-1: macrophage chemoattractant protein-1, MR: mendelian randomization, OR: odds ratio, PD: pathological pockets, PI: plaque index, SIR: standardized incidence rates, T1D: type 1 diabetes TFTs: thyroid function tests, TSH: thyroid-stimulating hormone, VPI: visible plaque index.

**Table 3 ijerph-21-00860-t003:** Studies concerning the common pathogenetic factors shared by periodontitis and thyroid disease.

Authors	Study Design	Study Sample	Parameters Evaluated	Results
Al-Hindawi S., 2017 [145]	Cross-sectional	30 hypothyroid subjects with periodontitis 30 hypothyroid subjects without periodontitis 30 healthy controls	Frequency of HLA	Increased frequency of HLA-DRB1*03 and *04 alleles in hypothyroid patients with or without periodontitis The increased frequency of the HLA-DRB1*08 allele in a healthy group Increased frequency of the HLA DRB1*03 allele was in patients with hypothyroidism and periodontitis
Liu W., 2019 [146]	Experimental laboratory-based	Periodontal ligament tissues from 34 subjects with periodontitis and 34 non-periodontitis subjects	Expression levels of PTCSC3 and TLR4 mRNA in periodontitis-affected PDLSCs. Effects of PTCSC3 and TLR4 Proliferation ability of periodontitis-affected PDLSCs	PTCSC3 was downregulated, TLR4 was upregulated in PD-affected PDLSCs, and PTCSC3 overexpression inhibited proliferation
Zeng Y., 2023 [147]	Observational cross-sectional	Human periodontal ligament tissue was obtained from extracted bicuspid teeth before orthodontic treatment.	Osteogenic differentiation parameters in human PDLSCs exposed to various concentrations of TSH: alkaline phosphatase activity, alizarin red staining for calcium nodules, and expression levels of osteogenic genes, namely, OPN, RUNX2, COL1A1, and OCN	TSH hindered osteogenic differentiation in periodontal ligament stem cells, as evidenced by reduced alkaline phosphatase activity, calcium nodule formation, and expression of OPN, RUNX2, COL1A1, and OCN
Zheng L., 2023 [148]	Observational cross-sectional	2.943 subjects after applying appropriate NHANES sample weights.	Thyroid function markers TSH, fT3, and fT4 levels, alongside oral microbiome diversity metrics such as species richness and evenness	Both clinical and subclinical hyperthyroidism are associated with reduced diversity of the oral microbiome. Anti-thyroid oxidase antibodies and hypothyroidism are associated with increased oral biodiversity
Sahar H. Al-Hindawi, 2019 [149]	Analytical cross-sectional	30 hypothyroid subjects with periodontitis 30 hypothyroid subjects without periodontitis 30 healthy subjects	Serum and salivary levels of IL-1β, IL-6, and TNF-α	Increased serum and salivary IL-1β levels in hypothyroid patients with and without periodontitis. Increased salivary IL-6 levels in hypothyroid patients with periodontitis

COL1A1: collagen type I alpha 1 chain, ft3: tri-iodiothyroxine, ft4: thyroxine, HLA: human leukocyte antigen, IL-1β: interleukine-1β, IL-6: interleukine-6, OCN: osteocalcin, OPN: osteopontin, PDLSCs: periodontal ligament stem cells, PTCSC3: candidate papillary thyroid carcinoma susceptibility gene 3, RUNX2: RUNX family transcription factor 2, TLR4: Toll-like receptor 4, TNF: tumor necrosis factor, TSH: thyroid-stimulating hormone.

**Table 4 ijerph-21-00860-t004:** Studies concerning the effect of periodontal treatment on thyroid disease and the effect of thyroxine replacement on periodontitis.

Authors	Study Design	Study Sample (n.)	Treatment	Follow-Up	Parameter Evaluated	Outcomes
Vallabhan C.G., 2020 [150]	Comparative interventional	Study group: 20 hypothyroid subjects with periodontitis Control group: 20 patients with periodontitis	SRP	4 weeks	PI, GI, CAL, PPD, IL-6, and TNF-α	Reduction of PI, GI, CAL, PPD, IL-6, and TNF-α in both group
Bhankhar R., 2017 [151]	Prospective interventional	Study group: 15 hypothyroid subjects under medication Control group: 15 healthy subjects	NSPT	3 months	OHI-S, PBI, PSR, CAL, and serum TSH levels	Improvements in OHI-S, PBI, PSR, and CAL Reduction in serum TSH levels after NSPT
Hwang I.M., 2014 [152]	Retrospective cohort	38.902 subjects with periodontitis and 77.804 comparison subjects	SRP and periodontal flap surgery	13 years	Cancer incidence, demographics, comorbidities, sex- and age-specific cancer risks, aHR, cancer site-specific incidence rates, and aHR	Reduced overall cancer (aHR = 0.77) but increased aHR for prostate and thyroid cancers (aHR = 1.54)
Yusubova, Sh.R., 2021 [153]	Prospective cohort	Study group: 13 subjects with thyroid dysfunction and periodontitis Comparison group: 11 healthy subjects Control group: 12 subjects undergoing various treatment methods	Hyaluronic acid in the study group, chlorhexidine in the comparison group, and curettage in the control group	Before and after treatment	Periodontal indices Biochemical parameters in oral fluid: MDA levels, SOD activity, catalase activity Concentration of IgA, IgG, and IgA in saliva	Improved periodontal health, decreased MDA, increased SOD, catalase, and enhanced salivary immunoglobulin level
Rahangdal S.I., Galgali S.R., 2018 [92]	Comparative cross-sectional	52 hypothyroid subjects on thyroxine therapy and 50 healthy controls	Thyroxine therapy	12 months	PI, BI, PPD CAL, radiographic mandibular cortical thickness, and panoramic mandibular index parameters	Higher PPD and CAL in hypothyroid patients on thyroxine therapy indicate an increased risk for periodontal destruction

aHR: adjusted hazard ratio, BD: bleeding index, BPO: bleeding of probing, CAL: clinical attachment level, MDA: malondialdehyde, NSPT: non-surgical periodontal treatment, Oral hygiene index simplified, PBI: papillary bleeding index, PI: plaque index, PPD: probing pocket depth, SRP: scaling and root planning, periodontal screening and recording index, SOD: superoxide dismutase, Ig: immunoglobulin, TSH: thyroid-stimulating hormone.

## Data Availability

No new data were created or analyzed in this study. Data sharing is not applicable to this article.

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
