# Peer review of "Bidirectional Association between Periodontitis and Thyroid Disease: A Scoping Review"

_ijerph, 2024, doi:10.3390/ijerph21070860_

Round 1
Reviewer 1 Report
Comments and Suggestions for Authors
The authors only included Open Access articles. That is an important bias because several findings are not probably recorded. They should comment on this limitation of the study.
On line 364 y 365, the word "Thyroid cancer" is repeated for different concept
Author Response
The authors only included Open Access articles. That is an important bias because several findings are not probably recorded. They should comment on this limitation of the study.
Thank you for your observation, unfortunately we have chosen to select only open access articles because many are paid. This observation has been mentioned among the limitations of the study (line 791, conclusions). Additions have been underlined in green, deletions are written in red.
On line 364 y 365, the word "Thyroid cancer" is repeated for different concept.
Thank you, it has been corrected.
Reviewer 2 Report
Comments and Suggestions for Authors
The current manuscript aims to study the relationship between periodontitis and thyroid diseases via synthesizing evidence from studies published within 2014-2024.
General comments
1. How did the authors select years of publication to be screened (2014-2024)? And why only open access? As this may cause bias in study selection
2. Could the authors describe further on "automatic" filters and what it meant by "off topic"?
3. Authors should consider revisiting the PRISMA website for the latest version of flowchart and please kindly include reference for this.
4. Similarly for risk of bias analysis - do include reference for this tool.
5. The discussion section needs to be extensively improved. Odds ratio was presented for some studies in text but not in Table summarizing all selected studies.
The current discussion section lacks of comparison and each paragraph is written as a summary of a study. Authors should consider segregating the studies into different categories before synthesizing evidence from different perspective.
Comments on the Quality of English LanguageThe current discussion section lacks of comparison and each paragraph is written as a summary of a study. Authors should consider segregating the studies into different categories before synthesizing evidence from different perspective.
Author Response
- How did the authors select years of publication to be screened (2014-2024)? And why only open access? As this may cause bias in study selection. Thank you for your observation, we have chosen to select articles from the last 10 years to have feedback on the most current and up-to-date results possible. Unfortunately, we have chosen to select only open access articles because many are paid. The observation about the open access has been mentioned as limitation of the study (line 791, conclusions).
- Could the authors describe further on "automatic" filters and what it meant by "off topic"? Thank you for your observation. When possible, the automatic filters of the electronic databases PubMed, Scopus, Web of Science, and Cochrane help us to simplify the search. The main automatic filter applied are the following: text availability, article type, publication date, species, article language. By OFF TOPIC we mean articles that do not respect our search criteria, that is: articles concerning the association between periodontitis and thyroid (hypothyroidism, hyperthyroidism, Hashimoto's thyroiditis, Graves' disease, thyroid nodules, and thyroid cancer).
- Authors should consider revisiting the PRISMA website for the latest version of flowchart and please kindly include reference for this. The paper has been reorganized as a scoping review (Preferred Reporting Items for Systematic Reviews and Meta-Analyses statement for re-porting scoping reviews (PRISMA-ScR) flow-chart.
- Similarly for risk of bias analysis - do include reference for this tool. References have been added to bias assessment (figure 2).
- The discussion section needs to be extensively improved. Odds ratio was presented for some studies in text but not in Table summarizing all selected studies. OR and aHR have been added in table 2 (Song et al 2021, Kim et al 2022, Chen et al 2023, Hwang I.M., 2014, Gao et al 2024).
The current discussion section lacks of comparison and each paragraph is written as a summary of a study. Authors should consider segregating the studies into different categories before synthesizing evidence from different perspective. The discussion has been improved by linking articles and moving some paragraphs.
Comments on the Quality of English Language: The current discussion section lacks of comparison and each paragraph is written as a summary of a study. Authors should consider segregating the studies into different categories before synthesizing evidence from different perspective. We have made changes to the English language. Additions have been underlined in green, deletions are written in red.
Reviewer 3 Report
Comments and Suggestions for Authors
The authors have comprehensively reviewed the link between periodontitis and thyroid dysfunction and tried to present their data as a systematic review. However, I like to highlight that systematic reviews have a more focussed question. The paper here is more of a Scoping review or literature review. The authors are trying to answer to broad question: 1) identifying the association between a specific condition and exposure to a risk factor of various types and 2)“Do patients with thyroid disease (P) have clinical and laboratory benefits (O) from periodontal treatment (I) than the control group (C)?” Though the former is well-established and the latter requires a more evidence-based approach. However, answering both questions together in a single review is over-ambitious and made the paper too lengthen. The results are poorly presented and not as per the PRISMA guidelines of a typical systematic review. The evidence on the effect of periodontal therapy on thryoid function requires a meta-analysis which is not presented.
All components of PRISMA are also not adequately written. For example, the search strategy is inappropriate and does not consider any synonyms of periodontitis. Apart from using the MESH term, a proper search string is required. Also, the search string for each database is needed along with date of search used should be mentioned as a supplementary file. Please elaborate on the screening and data extraction process.
The results are incomplete and do not follow the typical systematic review format. The data which should be there in the results are mentioned in the discussion making it too elaborate and not focused. However, I recommend reorganising the paper as a narrative review.
Please concise the conclusion and give only the key evidence. If the author plan to do a systematic review, I suggest meta-analysis to show the effect of periodontal therapy and thyroid should be added.
Comments on the Quality of English LanguageThe language can be impoved.
Author Response
The authors have comprehensively reviewed the link between periodontitis and thyroid dysfunction and tried to present their data as a systematic review. However, I like to highlight that systematic reviews have a more focussed question. The paper here is more of a Scoping review or literature review. The authors are trying to answer to broad question: 1) identifying the association between a specific condition and exposure to a risk factor of various types and 2)“Do patients with thyroid disease (P) have clinical and laboratory benefits (O) from periodontal treatment (I) than the control group (C)?” Though the former is well-established and the latter requires a more evidence-based approach. However, answering both questions together in a single review is over-ambitious and made the paper too lengthen. The results are poorly presented and not as per the PRISMA guidelines of a typical systematic review. The evidence on the effect of periodontal therapy on thryoid function requires a meta-analysis which is not presented. All components of PRISMA are also not adequately written. For example, the search strategy is inappropriate and does not consider any synonyms of periodontitis. Apart from using the MESH term, a proper search string is required. Also, the search string for each database is needed along with date of search used should be mentioned as a supplementary file. Please elaborate on the screening and data extraction process. The results are incomplete and do not follow the typical systematic review format. The data which should be there in the results are mentioned in the discussion making it too elaborate and not focused. Thank you, it has been modified.
However, I recommend reorganizing the paper as a narrative review. According to reviewer’s suggestion, the paper has been reorganized as a scoping review.
Please concise the conclusion and give only the key evidence. Thank you, it has been done.
If the author plan to do a systematic review, I suggest meta-analysis to show the effect of periodontal therapy and thyroid should be added. According to reviewer’s suggestion, the paper has been reorganized as a scoping review.
Comments on the Quality of English Language: The language can be impoved. We have made changes to the English language. Additions have been underlined in green, deletions are written in red.
Round 2
Reviewer 1 Report
Comments and Suggestions for Authors
I have reviewed the corrected versión and I think that authors included the requested corrections
Author Response
Reviewer 1
I have reviewed the corrected version and I think that authors included the requested corrections.
We thank the reviewer for suggesting changes to improve the quality of our article.
Reviewer 2 Report
Comments and Suggestions for Authors
The current manuscript has been revised slightly.
General comments:
1. The authors should list out what it means by "off-topic" articles in the exclusion criteria.
2. Please ensure to obtain copyright for images used in the manuscript.
3. Gene names should be italicized.
4. The information presented is scattered despite the authors have attempted to segregate into different factors. For instance, for genes related study, the authors are encouraged to present general pathways involved. The relationship of periodontitis and thyroid diseases can in fact be bidirectional based on what has been presented - but how does it occur via genetic network?
5. Oral microbiome has been touched lightly - how many articles have presented such information? What are the significance of the field based on evidence presented?
6. There are multiple layers to this article - the authors mentioned that thyroid diseases have been covered - hypothyroidism, hyperthyroidism, Hashimoto's thyroiditis, Graves' disease, thyroid nodules, and thyroid cancer. The authors are encouraged to segregate the information based on different diseases before dissecting to different factors like genetic factors, pro-inflammatory cytokines, oral microbiome etc.
Comments on the Quality of English LanguageMinor typo throughout text.
Author Response
Reviewer 2
The current manuscript has been revised slightly.
General comments:
- The authors should list out what it means by "off-topic" articles in the exclusion criteria.
We thank you for the suggestion. Changes relating to the second round of revisions have been highlighted in yellow, to distinguish them from those relating to the first round which are in green. The requested clarification is made explicit from line 178 to line 180.
- Please ensure to obtain copyright for images used in the manuscript.
The figure was created by us especially for this article. and there is no similar figure in literature. We signed the copyright form.
- Gene names should be italicized.
Done (line 309, line 552, line 554 and 555, line 575 and 576, line 579 and 580, line 588 and 589, line 850).
- The information presented is scattered despite the authors have attempted to segregate into different factors. For instance, for genes related study, the authors are encouraged to present general pathways involved. The relationship of periodontitis and thyroid diseases can in fact be bidirectional based on what has been presented - but how does it occur via genetic network? The changes indicated by the reviewer are reported in paragraph 4.1.3, entitled "Genetic factors" (lines 539-544, lines 557-561, lines 567-570, lines 593-596, line 560, lines 597-602).
- Oral microbiome has been touched lightly - how many articles have presented such information? What are the significance of the field based on evidence presented? Thank you for your suggestion. The part on the oral microbiome has been expanded from line 610 to line 629.
- There are multiple layers to this article - the authors mentioned that thyroid diseases have been covered - hypothyroidism, hyperthyroidism, Hashimoto's thyroiditis, Graves' disease, thyroid nodules, and thyroid cancer. The authors are encouraged to segregate the information based on different diseases before dissecting to different factors like genetic factors, pro-inflammatory cytokines, oral microbiome etc.
We would like to highlight that in many studies the different thyroid pathologies are associated in the study sample, therefore it would not be feasible for us to structure the discussion based entirely on the different thyroid pathology examined. In our opinion, it would be difficult to divide all the articles into different subparagraphs, because many of them mention multiple pathologies and cannot be separated. Nevertheless, we tried to segregate the information as suggested by the reviewer in paragraph 3.5 entitled “Synthesis of results”.
Reviewer 3 Report
Comments and Suggestions for Authors
The authors have improved the manuscript consisderably, However, I have a few queries:
1. The authors stated that the review is a scoping review, instead of a systematic review and have followed the " Systematic Reviews and Meta-Analyses statement for reporting scoping reviews (PRISMA-133 ScR) guidelines. However, Scoping reviews do not generally include the risk of BIAS analysis. Critical appraisal or risk of bias assessment is generally not recommended in scoping reviews because the aim is to map the available evidence rather than provide a synthesized and clinically meaningful answer to a question. For this reason, an assessment of methodological limitations or risk of bias of the evidence included within a scoping review is generally not performed (Ref: Peters, Micah D.J.1,2,3; Marnie, Casey1; Tricco, Andrea C.4,5,6; Pollock, Danielle7; Munn, Zachary7; Alexander, Lyndsay8,9; McInerney, Patricia10,11; Godfrey, Christina M.6,12; Khalil, Hanan13,14. Updated methodological guidance for the conduct of scoping reviews. JBI Evidence Synthesis 18(10):p 2119-2126, October 2020. | DOI: 10.11124/JBIES-20-00167 ).
the results section should be presented according to guidelines of the PRISMA-ScR checklist. The results section should be revised accordingly.
please note the heading of the scoping review for esults and discussion are as follows:
|
RESULTS |
|
Selection of sources of evidence |
|
Characteristics of sources of evidence |
|
Critical appraisal within sources of evidence |
|
Results of individual sources of evidence |
|
Synthesis of results |
|
DISCUSSION |
|
Summary of evidence |
|
Limitations |
|
Conclusions |
REF: https://www.prisma-statement.org/scoping
Comments on the Quality of English Language
Can be improved
Author Response
Reviewer 3
The authors have improved the manuscript considerably, However, I have a few queries:
- The authors stated that the review is a scoping review, instead of a systematic review and have followed the " Systematic Reviews and Meta-Analyses statement for reporting scoping reviews (PRISMA-133 ScR) guidelines. However, Scoping reviews do not generally include the risk of BIAS analysis. Critical appraisal or risk of bias assessment is generally not recommended in scoping reviews because the aim is to map the available evidence rather than provide a synthesized and clinically meaningful answer to a question. For this reason, an assessment of methodological limitations or risk of bias of the evidence included within a scoping review is generally not performed (Ref: Peters, Micah D.J.1,2,3; Marnie, Casey1; Tricco, Andrea C.4,5,6; Pollock, Danielle7; Munn, Zachary7; Alexander, Lyndsay8,9; McInerney, Patricia10,11; Godfrey, Christina M.6,12; Khalil, Hanan13,14. Updated methodological guidance for the conduct of scoping reviews. JBI Evidence Synthesis 18(10): p 2119-2126, October 2020. | DOI: 10.11124/JBIES-20-00167).
We thank you for the suggestion and appreciation for our effort in improving the article. Changes relating to the second round of revisions have been highlighted in yellow, to distinguish them from those relating to the first round which are in green. We have eliminated paragraph 3.3 (Quality Assessment and Risk of Bias of Included Articles) and figure 2 (Bias assessment) from the manuscript. Consequently, the numbering of the next figure (figure 3) has changed, which is now figure 2. Finally, we also included the reference indicated by the reviewer (reference number 142, 134 line).
The results section should be presented according to guidelines of the PRISMA-ScR checklist. The results section should be revised accordingly. please note the heading of the scoping review for results and discussion are as follows:
RESULTS
Selection of sources of evidence
Characteristics of sources of evidence
Critical appraisal within sources of evidence
Results of individual sources of evidence
Synthesis of results
DISCUSSION
Summary of evidence
Limitations
Conclusions
REF: https://www.prisma-statement.org/scoping
The discussion has been edited as recommended by the reviewer. The paragraph 4.2. entitled “limitations of the study” was created, moving the part from line 861 to line 868 that was previously included in the conclusion. Additionally, the “summary of evidence” paragraph was included. Consequently, the numbering of the following paragraphs has been corrected.